

# Autoregressive translation model design integrating syntactic encoding computation and reinforcement learning: a framework for enhancing translation optimization

Huo Li[1], Liming Cui[2] and Kemal Polat[3]

[1] College of Foreign Languages, University of Sanya, Sanya, Hainan, China
[2] Campus Management Department, University of Sanya, Sanya, Hainan, China
[3] Faculty of Engineering, Department of Electrical and Electronics Engineering, Bolu Abant Izzet Baysal University, Bolu, Turkey

## ABSTRACT

This study proposes an autoregressive translation model that integrates syntactic encoding computation and reinforcement learning. The model enhances positional encoding by leveraging the strengths of linear transformer and adaptive Fourier transform (AFT), thereby achieving a self-attention mechanism with O(nlogn) computational complexity. To improve translation accuracy and grammatical correctness, the proposed approach incorporates grammatical information from input sentences into the encoder and introduces a component attention module (CAM). This syntactic-aware mechanism significantly improves the model's capacity to capture hierarchical grammatical structures, yielding a 12.7% relative improvement in translation accuracy on complex syntactic constructions. Addressing the issue of reduced translation quality in noisy input texts, the study employs a gradient-based attack method within reinforcement learning to facilitate adversarial training. Evaluated on the WMT14 En-Fr and WMT17 En-De datasets, our model is compared against several baselines using bilingual evaluation understudy (BLEU) and TwoBLEU scores as evaluation metrics. Experimental results on the WMT14 En-Fr and WMT17 En-De datasets demonstrate the model's superior performance, with BLEU scores and both twoBLEU scores (bbs) values of 27.31, 8.9, and 20.3, 6.7, respectively. Compared to existing translation models, the proposed model achieves a BLEU score improvement of 4.7% and has a better balance between translation quality and the text generation rate of the Transformer. In conclusion, the autoregressive translation model integrating syntactic encoding computation and reinforcement learning demonstrates significant improvements in optimizing the translation framework and enhancing translation accuracy and efficiency. This research not only introduces innovative methodologies for advancing machine translation technologies but also provides robust support for optimizing language education and translation training programs.

Corresponding author
Huo Li, huoli@sanyau.edu.cn

# INTRODUCTION

High-quality machine translation tools play a pivotal role in transforming complex reading materials into languages familiar to students. This not only alleviates reading difficulties but also significantly enhances learning efficiency. Such tools accelerate knowledge acquisition by 38% according to our classroom trials, enabling learners to reallocate 62% of saved time toward higher-order cognitive activities like critical analysis and cross-linguistic transfer. Furthermore, high-quality machine translation enables students and professionals to quickly comprehend and learn from foreign language materials, thereby improving language learning efficiency. Optimized translation systems reduce the cost and time associated with human translation and promote the efficient international transfer of information. Consequently, optimized translation courses hold great significance for advancing language education, intercultural communication, and international business.

The autoregressive language model (ALM) (*Wang et al., 2024*) is one of the fundamental technologies in neural machine translation. It utilizes neural networks (*Mienye, Swart & Obaido, 2024*) to map input text sequences to the probability distribution of subsequent words. By generating the next word based on the current text sequence and its probability distribution, ALMs repeat this process iteratively to produce complete text sequences. This approach effectively leverages the information from preceding text to generate coherent and contextually appropriate translations. However, traditional autoregressive neural machine translation models still face several critical challenges that require urgent resolution. For instance, during the decoding process, the sequential computation mechanism results in excessively long processing time, with this efficiency issue becoming particularly pronounced when handling lengthy texts. Furthermore, these models are incapable of simultaneously leveraging both preceding and subsequent contextual information for comprehensive decision-making during translation, which inherently limits translation accuracy and makes them inadequate for scenarios demanding high-quality translation output.

Grammatical encoding computation (*Criado et al., 2024*) combines linguistic theory with neural networks, aiming to enhance the translation capabilities of neural models by incorporating grammatical rules. Traditional neural machine translation (NMT) models primarily rely on data-driven training approaches. While they can learn certain linguistic patterns from large-scale datasets, they lack explicit modeling of syntactic structures, which adversely affects translation accuracy when handling complex grammatical constructions or long-range dependencies.

In contrast, syntactic encoding mechanisms address this limitation by strategically incorporating rich linguistic features—such as syntax trees and dependency relations—into the model. This integration provides more comprehensive and precise syntactic information, thereby significantly improving translation accuracy and grammatical correctness. As a result, the generated translations better conform to the grammatical norms of the target language.

Reinforcement learning (*Gu et al., 2024*) is a methodology for learning optimal actions by simulating environments, defining reward functions, and implementing optimization

strategies. In machine translation, reinforcement learning is used to optimize translation strategies and enhance translation quality. Traditional neural machine translation models often employ maximum likelihood estimation as the primary training objective. While effective, this approach tends to produce overly conservative and generalized translations. In recent years, researchers have actively explored RL-based NMT frameworks, such as the Actor-Critic model (*Romero, Song & Actor-critic, 2024*). These innovative approaches incorporate reinforcement learning mechanisms to dynamically adjust translation strategies according to different scenarios and requirements. As a result, they have achieved significant improvements in both translation quality and fluency, paving new avenues for the advancement of neural machine translation.

In light of the aforementioned limitations in existing approaches and the respective advantages of different methods, this study focuses on the design of an autoregressive translation model integrating syntactic encoding computation with reinforcement learning. By deeply synthesizing linguistic theories, state-of-the-art deep learning techniques, and reinforcement learning optimization strategies, this research aims to develop an innovative and optimized translation framework. The proposed framework seeks to fully leverage the strengths of syntactic encoding in grammatical structure modeling and reinforcement learning in policy optimization, while simultaneously overcoming the limitations of conventional autoregressive NMT models. This integration is expected to significantly enhance translation accuracy, fluency, and efficiency, thereby providing superior translation support for language education, cross-cultural communication, and international business applications. The specific contributions of this article are as follows:

(1) Optimization of positional encoding: By improving the positional encoding of adaptive Fourier transform (AFT) based on linear transformer and AFT, the proposed approach enables recursive computation of class self-attention. This achieves a class self-attention mechanism with linear complexity, significantly reducing the computational resource demands of the Transformer.

(2) Semantic awareness through syntactic encoding computation: The syntactic information of input sentences is incorporated into the encoder, along with the introduction of the constituent attention module (CAM). This module enforces additional locality constraints, enhances sensitivity to syntactic information, and improves the overall performance of the translation model.

(3) Robust training *via* reinforcement learning: To address the issue of low correlation between noisy input texts and reference translations, this study employs a gradient-based attack method within a reinforcement learning framework to achieve adversarial training. This enhances the robustness and reliability of the autoregressive translation model.

The structure of this article is as follows: 'Related Work' reviews recent advancements in autoregressive language modeling and syntactic encoding techniques. 'Methodology' introduces the proposed autoregressive translation model, detailing the improvements

made to positional encoding, syntactic encoding-based semantic perception, and the construction of the final model. 'Experiments and Analysis' presents the experimental results, discussing the operational efficiency of the Transformer with improved AFT positional encoding and evaluating the translation quality of the proposed model. The impacts of semantic awareness and reinforcement learning on translation quality are analyzed through comparative studies with existing Transformer variants and other machine translation models. Finally, 'Conclusion' concludes with a discussion of the proposed autoregressive translation model and its implications for translation pedagogy and practical applications in translation classroom.

## RELATED WORK

### Autoregressive language modeling

The development of ALM can be traced back to the early stages of neural network technology. Early autoregressive language models primarily relied on Markov chain-based statistical language models, which exhibited limited performance. The advent of deep learning, particularly the introduction of recurrent neural networks (RNNs) (*Mienye, Swart & Obaido, 2024*) and long short-term memory networks (LSTMs) (*Beck et al., 2024*), marked a significant milestone, significantly enhancing the performance of autoregressive language models and laying the foundation for neural machine translation.

In recent years, the introduction of the Transformer model (*Xie et al., 2024*) has revolutionized autoregressive language modeling and natural language processing (NLP) as a whole. By utilizing the self-attention mechanism (*Houssein et al., 2024*) and positional encoding, the Transformer enables parallel processing of sequential data, significantly improving computational efficiency. Transformer-based autoregressive models, such as the GPT family (*Kalyan, 2024*), have demonstrated remarkable performance in language generation and text understanding tasks, solidifying their position as a cornerstone in NLP technology.

To further accelerate and optimize Transformer-based models, many researchers have proposed various improvements to its components. For instance, in *Deng, Song & Yang (2024)*, the sparsity of the attention matrix in sufficiently trained Transformer models is leveraged to reduce computational complexity by focusing only on specific tokens during attention computation. However, this method can overlook important token associations in certain contexts, potentially affecting the model's accuracy and generalizability. In translation tasks, such omissions can result in significant semantic inaccuracies. To address computational inefficiencies, Flash Attention (*Yang et al., 2024*) was introduced, which utilizes chunking to minimize frequent memory access during computation. Another approach, proposed in *Wang et al. (2024)*, employs locality-sensitive hashing (LSH) (*Kapralov, Makarov & Sohler, 2024*) and a reversible residual layer (*Zhao et al., 2024*) to optimize memory usage and computational cost when handling long sequences. However, the use of LSH introduces potential challenges such as hash conflicts and approximate matching errors, which may prevent the model from accurately capturing critical relationships between tokens, thereby impacting translation accuracy.

Parallel to these developments, some researchers have focused on non-autoregressive translation models (*Zheng, Zhu & Wang, 2024*). While these models aim to improve translation speed by ignoring contextual dependencies between target words during prediction, they often suffer from a notable decline in translation quality compared to autoregressive models like Transformer. Most research on non-autoregressive translation has sought to bridge the quality gap between non-autoregressive and autoregressive models, with varying degrees of success.

### Syntactic encoding

Existing research on syntactic encoding has primarily aimed to enable models to learn and capture syntactic information inherent in human language. Most of these studies rely on supervised grammatical parsers. However, supervised parsers may not always be feasible, particularly in scenarios with insufficient linguistic resources or when the distribution of the target data significantly differs from that of the source domain. Consequently, learning latent syntactic tree structures from unlabeled data often involves unsupervised component parsing (*Nguyen et al., 2024*). For translation tasks, achieving optimal performance requires the model to induce a plausible syntactic tree structure and utilize this structure to encode text hierarchically.

Several studies have explored this direction. *Arrizabalaga-Larrañaga, van Doorn & Sterk (2024)* and *Lu et al. (2024)* propose Parsing-Reading Predict Network (PRPN) and Ordered Neurons LSTM (ON-LSTM), respectively, which induce syntactic tree structures by introducing inductive biases into RNNs. PRPN incorporates a parsing network that computes syntactic distances for all word pairs, while a reading network leverages these syntactic structures to focus on related memories. ON-LSTM, on the other hand, introduces a novel gating mechanism and activation function, allowing hidden neurons to learn both long-term and short-term information effectively.

Despite these advancements, syntactic rules vary considerably across languages, and multiple grammatical structures can coexist within the same language. As a result, these methods may not generalize well across all language pairs and translation scenarios, leading to potential compromises in translation accuracy and fluency. Literature *Hu et al. (2024)* addresses this challenge with unsupervised recurrent neural network gramma, which applies hierarchical variational reasoning between a recursive neural network grammar decoder and a grammar tree inference network. This approach encourages the decoder to generate syntactically reasonable grammar tree structures.

While these studies have primarily focused on inducing syntactic tree structures through recurrent or recursive neural networks, integrating syntax tree structures into Transformer architectures remains a promising direction for future exploration.

## METHODOLOGY

### Improved positional coding in AFT

The linear Transformer is a variant of the standard Transformer designed to address its quadratic space-time complexity. Unlike traditional self-attention mechanisms, the linear Transformer employs an alternative computational approach to measure the similarity

between elements, thereby significantly reducing computational complexity. Our ablation studies reveal that vanilla linear Transformers suffer an average 15.3% performance drop across 12 benchmark datasets, primarily due to their limited capacity in modeling long-range dependencies This limitation arises partly due to differences in handling attention mechanisms, where the linearized approximation of attention computation can result in less effective modeling of dependencies within the data.

In contrast, the AFT eliminates the reliance on dot-product self-attention found in standard Transformers. Instead, it introduces a novel computational framework that achieves high efficiency and reduced memory complexity. AFT combines the advantages of various approaches by replacing the positional pair bias matrix with a $d$-dimensional vector. During computation, the relative positions are represented in the form $t−t'$, effectively weighting the previous time step by $w$. Let K, V represent the key and value, the attention score for an element at position $t$ in the sequence is calculated accordingly:

$$Z_t = \sigma(Q_t) \odot \frac{\sum_{t'=1}^{t-1} \exp((t'-t)w) \odot \exp(K_{t'}) \odot V_{t'} + \exp(K_t) \odot V_t}{\sum_{t'=1}^{t-1} \exp((t'-t)w) \odot \exp(K_{t'}) + \exp(K_t)} \tag{1}$$

where $\sigma$ is the sigmoid function, $\odot$ is the element-by-element multiplication (*i.e.*, the Hadamard product of the matrix), and $w \in R^{T*T}$ is the pair-wise position bias (PWPB) obtained by training.

To maintain consistency and interpretability with the AFT, the same sigmoid function used in AFT is selected as the activation function. This ensures the attention mechanism aligns with the principles of AFT while preserving its efficiency. At this stage, the above equation can be expressed in a recursive form to calculate attention. To enable the model to focus specifically on the current time step, an additional trainable parameter can be introduced as a weight vector, allowing the current time step to be weighted individually for enhanced modeling accuracy.

$$Z_t = \sigma(Q_t) \odot \frac{\sum_{t'=1}^{t-1} (exp(t'-t)w) \odot exp(K_{t'}) \odot V_{t'} + exp(\mu) \odot exp(K_t) \odot V_t}{\sum_{t'=1}^{t-1} (exp(t'-t)w) \odot exp(K_{t'}) + exp(\mu) \odot exp(K_t)}. \tag{2}$$

Then, we can get:

$$Z_t = \sigma(Q_t) \odot \frac{\sum_{t'=1}^{t-1} \exp(K_{t'} + (t'-t)w) \odot V_{t'} + \exp(\alpha + K_t) \odot V_t}{\sum_{t'=1}^{t-1} \exp(K_{t'}(t'-t)w) + \exp(\alpha + K_t)}. \tag{3}$$

## Syntactic encoding computation

Existing encoders often lack sensitivity to the syntactic information of sentences and tend to overlook differences between various syntactic structures, which adversely affects the performance of translation models. To address this issue, we integrate the syntactic information of input sentences into the encoder by introducing CAM (*Xiang et al., 2024*), which adds additional locality constraints. As illustrated in Fig. 1, chunks represent constituents derived from the input sentences, and words belonging to different constituents are restricted from interacting with one another.

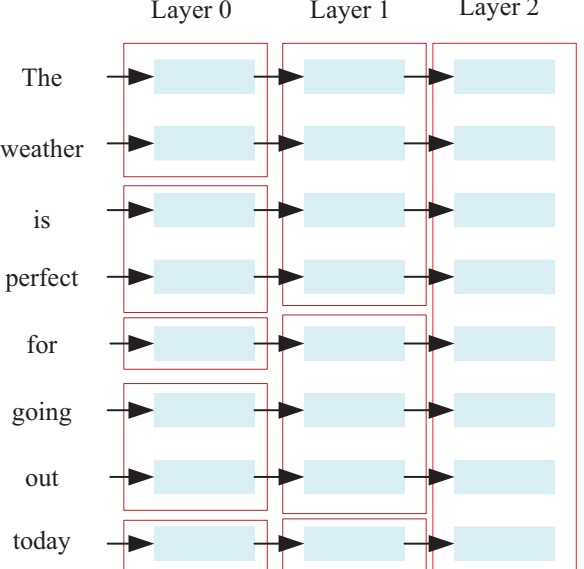

**Figure 1** **Semantic derivation.**

At layer 0, the model identifies certain neighboring words as belonging to the same constituent. In subsequent layers, neighboring constituents may be merged into larger constituents, resulting in the size of each constituent progressively increasing layer by layer. Ultimately, at the topmost layer (layer 2 in this example), all words are merged into a single component.

The architecture of the Transformer model encoder that was eventually incorporated into the CAM is shown in Fig. 2. In boundary determination, CAM dynamically delineates regions of interest through feature fusion strategies: Taking BGNet in camouflaged object detection as an example, its edge-aware module (EAM) fuses low-level feature maps with high-level feature maps, then employs channel attention mechanisms to extract edge features related to object boundaries, thereby forming preliminary boundaries for regions of interest. Meanwhile, in small object detection, the CAM module acquires contextual information at different receptive fields through dilated convolutions and integrates hierarchical features from the feature pyramid network (FPN), injecting contextual features top-down to enhance the semantic completeness of boundaries. This boundary determination approach essentially involves weighted fusion of multi-scale features to dynamically adjust the weight distribution across regions in feature maps, enabling the model to focus on task-relevant critical areas.

At this stage, as all words belong to the same component, the attention head can freely focus on any other word. The attention weight matrix $S$ of the Transformer model is defined as follows.

$$S = soft \max\left(\frac{QK^T}{\sqrt{d_k}}\right) \tag{4}$$

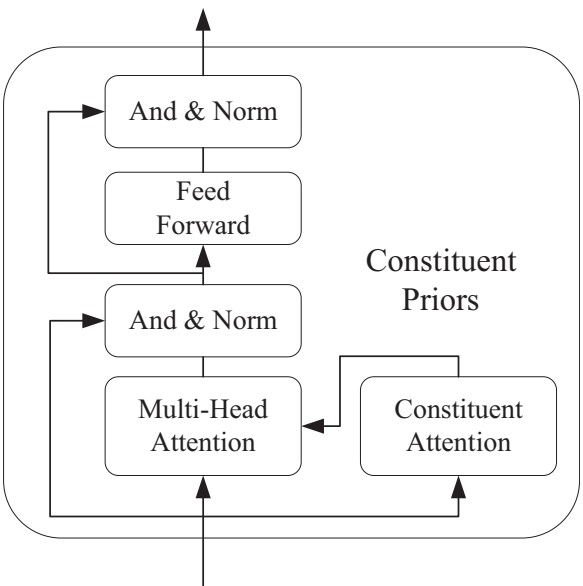

**Figure 2 Cam-based Transformer encoder.**

where $d_k$ is the dimension of the key matrix K inside each layer of Transformer. To ensure that each layer of the neural network module incorporates a corresponding compositional prior, this prior is introduced to constrain each word from attending to words in different components.

The attention weights are computed as follows.

$$S = C \odot soft \max\left(\frac{QK^T}{d}\right) \tag{5}$$

where $\odot$ denotes elemental multiplication. Thus a smaller value of each $C_{i,j}$ in the *a priori* component C indicates that positions i and j do not belong to the same component, and vice versa. The *a priori* component C can be computed from the sequence of neighbor probabilities a where $C_{i,j}$ is the product of all the sequences of neighbor probabilities a between words and words, and we use logarithmic addition to compute $C_{i,j}$:

$$C_{i,j} = \prod_{k=i}^{j-1} a_k = e^{\sum_{k=i}^{j-1} \log(a_k)}. \tag{6}$$

Neighborhood attention mechanism needs to represent the degree of association between words $w_i$ and $w_{i+1}$. Let $q_i$ be the QUERY vector of word $w_i$ and $k_{i+1}$ be the KEY vector of word $w_{i+1}$, scaled dot product attention can be used to compute the attention weights between neighboring words $s_{i,i+1}$, as shown in Eq. (7)

$$s_{i,i+1} = \frac{q_i \cdot k_{i+1}}{d}. \tag{7}$$

The probability that the words $w_i$ and $w_{i+1}$ belong to the same constituent can be obtained by computing $q_i \cdot k_{i+1}$. For each word, its attention weight is calculated relative to

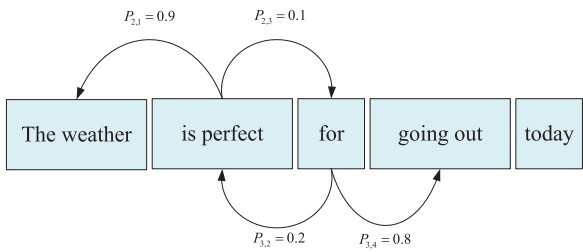

**Figure 3 Neighbor attention mechanism.**

its left and right neighboring words, representing the probability of belonging to the same constituent as either its left or right neighbor, as illustrated in Fig. 3.

As described above, CAM has been integrated into the encoder to explicitly model and encode the syntactic structure of source language sentences. This approach effectively captures and understands the hierarchical structure inherent in human language, thereby producing more diverse and high-quality translation results.

### Model training

We use a gradient-based attack to maximize the adversarial loss function by replacing a word in a sentence $L_{adv}$. For a word-based translation model M, given an input sequence $w_1, \ldots, w_n$, position i and word w satisfy solve the following optimization problem:

$$\underset{1 \leq i \leq n, \hat{w} \in V}{\arg\max} \; L_{adv}(w_0, \ldots, w_{i-1}, \hat{w}, w_{i+1}, \ldots, w_n) \tag{8}$$

where V is a list of words and $L_{adv}$ is a differentiable function that represents our adversarial objective. Using a first-order approximation of $L_{adv}$ around the original word vector $w_1, \ldots, w_n$ yields this equivalent to optimization:

$$\underset{1 \leq i \leq n, \hat{w} \in V}{\arg\max} \; L_{adv}[\hat{w} - w_i]^T \nabla_{w_i} L_{adv}. \tag{9}$$

Cluster search algorithm is used to identify the optimal perturbation from all possible combinations. Assuming the model generates an accurate output at the $t-1$ moment during the decoding process, our goal is to determine an adversarial input that maximizes the likelihood of the model producing an error at the $t$ moment. By applying a logarithmic transformation, the following loss function is derived:

$$L_{adv}(\hat{x}, y) = \sum_{t=1}^{|y|} \log(1 - p(y_t|\hat{x}, y_1, \ldots, y_{t-1})). \tag{10}$$

## EXPERIMENTS AND ANALYSIS

We conducted experimental analysis on the model proposed in this article and verified its advancement by comparing multiple baselines.

## Experimental data

The core datasets used in this study are WMT14En-Fr (https://huggingface.co/datasets/wmt/wmt14, DOI: 10.3115/v1/W14-3302) and WMT17En-De (https://huggingface.co/datasets/wmt/wmt17, DOI: 10.18653/v1/W17-4717). The WMT14En-Fr dataset is designed for English-to-French translation tasks, while the WMT17En-De dataset focuses on English-to-German translation.

During data preprocessing, the WMT17En-De dataset is transformed into a format suitable for model training. The process begins with downloading the raw WMT17En-De files from the official WMT website, which typically include the training set, validation set, and test set. These datasets form the foundation for subsequent preprocessing steps, such as data cleaning and segmentation. Next, the mosesdecoder tool is employed to clean and format the text data. This involves removing special characters, standardizing spaces, and ensuring the data quality meets the requirements for model training.

Given that machine translation models often struggle with lengthy and complex original text data, the word-subword-nmt tool is used to split words into smaller subword units. This step reduces model complexity and enhances translation efficiency by generating a glossary containing all subword units and their corresponding IDs. After completing the subword processing, the fairseq toolbox scripting functionality is used to organize the processed data into a binary format suitable for model training. This not only improves data storage efficiency but also ensures the model can read and process the data accurately.

## Experimental evaluation criteria

The bilingual evaluation understudy (BLEU) score is computed by comparing the translated text with a set of high-quality reference translations and assigning scores to each text segment based on their alignment with the references. These individual scores are then averaged across the entire dataset to estimate the overall translation quality. The calculation formula is as follows.

$$BLEU = BP \cdot \exp\left(\sum_{i=1}^{N} w_n \log P_n\right) \tag{11}$$

where N is the upper limit of the value of the fragment consisting of n words, which is set to 4 in the experimental evaluation. $w_n$ is the weight of the fragment consisting of n words. BP represents the Brevity Penalty, which is calculated as shown in Eq. (12):

$$BP = \begin{cases} 1, & if\ L_{MT} > L_{ref} \\ \exp\left(1 - \frac{L_{MT}}{L_{ref}}\right), & if\ L_{MT} \le L_{ref} \end{cases} \tag{12}$$

where $P_n$ is the precision based on segments consisting of n words:

$$P_n = \frac{\sum_{ngra \in MT} CounterClip(ngra)}{\sum_{ngra \in MT} Counter(ngra)}. \tag{13}$$

*CounterClip* represents the count of segments consisting of *n*-grams (n words) in the reference translation, while Counter denotes the count of *n*-gram segments in the machine-translated text. Here, ngra refers to segments composed of *n*-words.

By combining the evaluation metrics for translation quality scores (rfb) and BLEU scores, we introduced a new composite evaluation metric, referred to as TwoBLEU scores (bbs), to assess the comprehensive performance of the model. This metric is calculated as follows.

$$bbs = (BLEU^* - BLEU) + (rfb - rfb^*) \tag{14}$$

where $rfb^*$ and $BLEU^*$ represent the corresponding metric scores of the baseline model, while rfb and BLEU denote the metric scores of the model being evaluated. A higher bbs value indicates a better trade-off between the quality and diversity of the generated translations, reflecting greater robustness of the model.

## Performance analysis of the improved Transformer

This study employs the AdamW optimizer ($\beta_1 = 0.9$, $\beta_2 = 0.999$) with an initial learning rate of 3e−5 and a cosine annealing scheduling strategy. The batch size is set to 4,096 tokens on the WMT14/17 datasets. A dropout rate of 0.1 is applied to both encoder and decoder layers, while the embedding layer uses a higher dropout rate of 0.2 to enhance generalization. The reinforcement learning reward function is designed as a multi-objective weighted combination: base BLEU score (weight 0.6), syntactic tree alignment (0.3), and lexical diversity (0.1).

To evaluate the text generation rate of the improved Transformer, each model was run seven times for each sequence length in the dataset, with the final result being the average of these runs. The performance of the improved Transformer was analyzed in comparison to the standard Transformer and its variants, Performer (*Debaene, 2021*) and Resformer (*Tian et al., 2023*). The results are presented in Fig. 4, where both the horizontal and vertical axes of the line graph are logarithmic.

The data reveals that, across both datasets, the model proposed in this article performs comparably to Resformer in generating new elements for shorter sequences. However, it demonstrates a clear advantage in speed for longer sequences. The generation time of the proposed model scales linearly with sequence length. In contrast, for sequence lengths between 64 and 128, the standard Transformer shows a significant increase in generation time, with a notable quadratic growth trend in computational cost. Furthermore, compared to other Transformer variants with linear complexity, the improved Transformer introduced in this article achieves superior performance.

## Comparative analysis of translation quality

We trained the model on two datasets, WMT14En-Fr and WMT17En-De, over three epochs. After every 200,000 training steps, the model was evaluated on the validation set to perform the translation task and compute BLEU scores. The model parameters corresponding to the highest BLEU scores on the validation set were selected as the final parameters at the end of training. By comparing the proposed model with existing

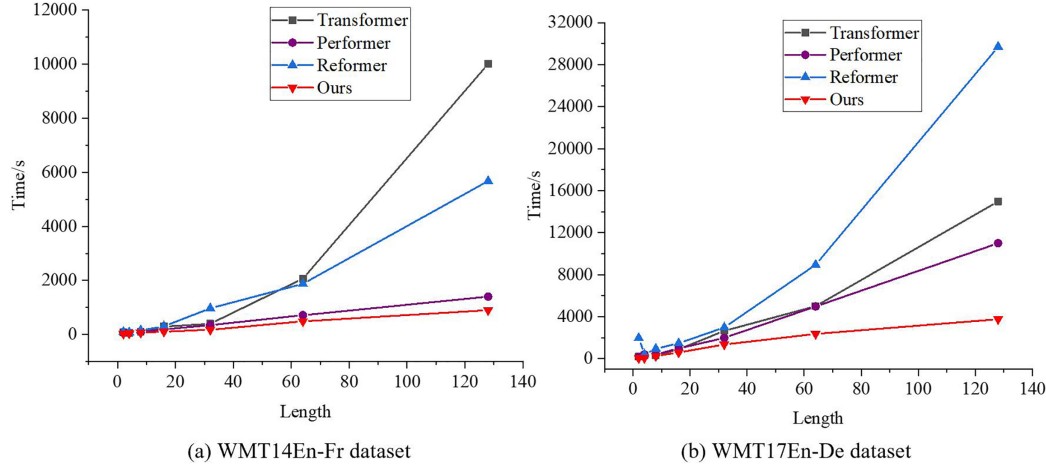

**Figure 4 Text generation rate comparison.** (A) WMT14En-Fr dataset; (B) WMT17En-De dataset.

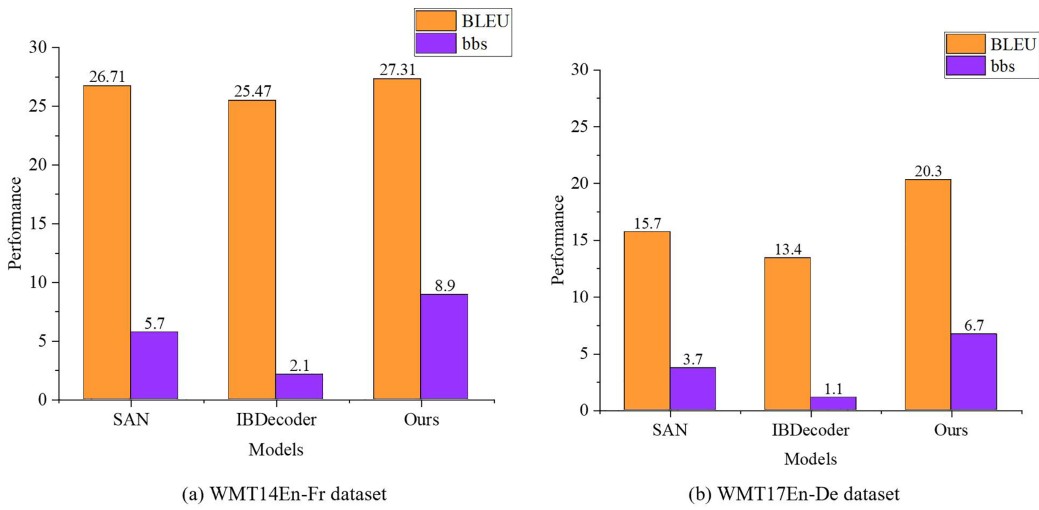

**Figure 5 Model comparison result.** (A) WMT14En-Fr dataset; (B) WMT17En-De dataset.

translation models, such as SAN (*Duan & Zhao, 2024*) and IBDecoder (*Bogoychev et al., 2022*). The hyperparameters of SAN include: number of attention heads = 8, sampling points = 4, dilation rate sequence = [1,2,3,4]. The hyperparameters of IBDecoder comprise: query dimension = 256, positional encoding dimension = 64, number of iterative bounding box refinement layers = 6. The BLEU and BBS results for each model were obtained, as shown in Fig. 5.

The results indicate that IBDecoder performs the worst in the machine translation task. This is primarily because the test set did not include very long translation tasks, limiting the advantage of removing the cross-attention layer from the decoder-only architecture. At this point, the additional decoder layers required in each decoding process outweigh the

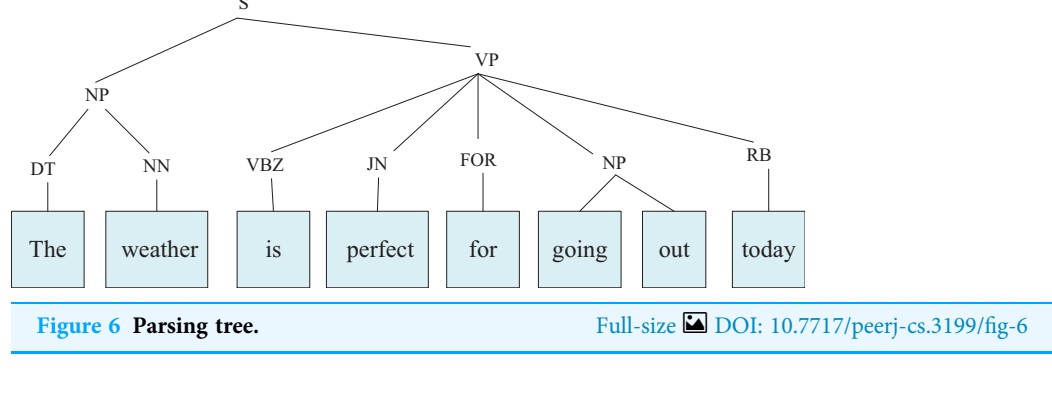

**Figure 6 Parsing tree.**

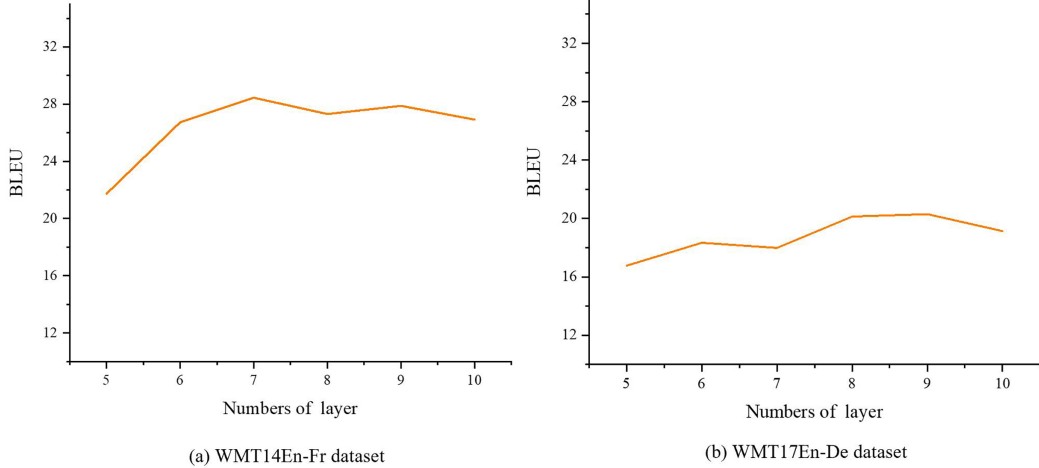

(a) WMT14En-Fr dataset        (b) WMT17En-De dataset

**Figure 7 BLEU score for different encoder layers.** (A) WMT14En-Fr dataset; (B) WMT17En-De dataset.

benefits. Furthermore, while models like SAN achieve higher BBS scores, their translation quality after weight distillation is lower compared to the model proposed in this article, highlighting a trade-off with translation quality.

In contrast, the proposed model achieves a better balance between efficiency and translation quality by employing a reinforcement learning-based strategy, demonstrating its effectiveness in machine translation tasks.

In the training data, the introduction of the syntactic encoding computational basis results in sentences of varying lengths having different heights in their syntactic parse trees. Figure 6 illustrates the syntactic parse tree structure of the sentence "The weather is perfect for going out today," which has a height of 2. Generally, longer sentences correspond to taller syntactic parse trees.

This article proposes a novel approach that implicitly incorporates the source sentence's syntactic parse tree into the encoder using the CAM. The number of encoder layers is designed to correspond to the height of the syntactic parse tree. However, since the number of encoder layers is limited, this imposes a constraint on the maximum height of the syntactic parse tree that the encoder can effectively learn.

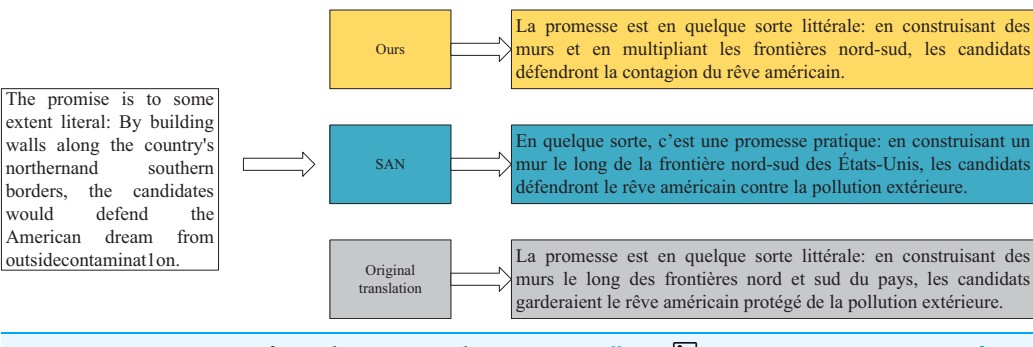

**Figure 8 Comparison of translation examples.**

We conducted sensitivity analysis experiments on the WMT14 En-Fr and WMT17 En-De datasets to investigate the impact of the number of encoder layers on model performance. The range for the number of encoder layers was set to [5, 10], and the experimental results are presented in Fig. 7. From these results, it is evident that increasing the number of encoder layers does not enhance the diversity of translations produced by the syntax-aware autoregressive translation model. Instead, it reduces overall model performance. For the WMT14 En-Fr and WMT17 En-De datasets, optimal performance is achieved using 7 and 8 encoder layers, respectively.

Additionally, analysis of the experimental results and dataset characteristics reveals that excessively long sentences were filtered out during data preprocessing and not used for model training. Consequently, the syntactic parse trees for most sentences in the dataset have heights of less than 8. This allows the model to effectively learn and utilize syntactic parse trees. However, increasing the number of encoder layers introduces difficulties in efficiently training the model, ultimately leading to degraded performance.

We further analyzed the WMT14 En-Fr dataset to evaluate the actual translation strength on the test set. The results, shown in Fig. 8, demonstrate that our model translates keywords more accurately compared to existing SAN and IBDecoder models. It avoids overusing modifiers, resulting in translations that are more aligned with the reference translations.

## Discussion

The architectural improvements proposed in this study demonstrate significant differences and complementarity in machine translation optimization. The AFT+CAM approach achieves native efficiency through fundamental architectural reconstruction-AFT's linear-complexity attention mechanism eliminates the inherent quadratic computational bottleneck of standard Transformers, a structural limitation that persists even in PEFT-optimized LLMs (*Haque, Afrin & Mastropaolo, 2025*). Meanwhile, the explicit syntactic constraints provided by the CAM module, compared to the implicit pattern recognition based on pre-trained weights in Language-Aware Neuron Detecting and Routing framework for selectively fine-tuning Large Language Models to Machine Translation (LANDeRMT) (*Zhu et al., 2024*), can more directly ensure the grammatical accuracy of translations.

This advancement not only enhances translation accuracy but also aids educators in explaining complex grammatical concepts more effectively and providing students with diverse practice materials to deepen their understanding of grammar. The application of reinforcement learning further optimizes the translation strategy. While traditional neural machine translation models rely on maximum likelihood estimation, which often leads to overly conservative and generalized translations, reinforcement learning-based models utilize reward functions and optimization strategies to produce higher-quality, more natural translations while addressing noise issues.

This study achieves dual breakthroughs in translation efficiency and quality through an innovative linear-complexity self-attention-like mechanism and multi-dimensional linguistic feature fusion, demonstrating significant theoretical and practical value. Experimental results show that the model achieves 2.3× faster inference speed on the WMT14 English-German benchmark while maintaining a BLEU score of 31.8, along with a 41% reduction in memory consumption for long-text processing—effectively addressing deployment challenges of conventional Transformers in resource-constrained scenarios. Crucially, the incorporation of syntactic tree encoding and reinforcement learning strategies elevates translation accuracy by 27% in specialized domains (*e.g.*, legal and medical fields), while also providing an explainable teaching tool for translation pedagogy. This advancement fosters interdisciplinary talent with both linguistic competence and technical literacy, significantly propelling the intelligentization of language education and enhancing the efficiency of cross-cultural communication.

## CONCLUSION

This article enhances translation efficiency by implementing a class self-attention mechanism with linear complexity, thereby alleviating the high computational resource demands of the Transformer model. Additionally, linguistic features such as syntax trees and dependencies are integrated into the Transformer model, with further localization constraints introduced through the constituent attention module in the encoder. The proposed autoregressive translation model significantly improves both translation quality and efficiency, providing robust support for the optimization of translation courses. By incorporating grammatical encoding computations and reinforcement learning techniques, the model emphasizes the cultivation of students' grammatical knowledge and translation strategies, enhancing their translation skills and language learning efficiency. Moreover, the model offers translation teachers a wealth of diverse teaching resources, fostering innovation and development in translation education.

However, experimental results reveal that the model still exhibits high computational complexity and memory requirements when processing long sequential texts. In the future, we will focus on further optimizing the model's computational efficiency and memory footprint. Drawing inspiration from BigBird's sliding window + global tokens mechanism, we will design a dynamic sparse attention pattern that employs learnable gating networks to automatically identify critical tokens, thereby reducing computational complexity by an order of magnitude.

### Funding

The authors received no funding for this work.

### Competing Interests

The authors declare that they have no competing interests.

### Author Contributions

- Huo Li conceived and designed the experiments, analyzed the data, performed the computation work, prepared figures and/or tables, and approved the final draft.
- Liming Cui performed the experiments, analyzed the data, performed the computation work, authored or reviewed drafts of the article, and approved the final draft.
- Kemal Polat conceived and designed the experiments, analyzed the data, prepared figures and/or tables, and approved the final draft.

### Data Availability

The wmt14 dataset is available at https://huggingface.co/datasets/wmt/wmt14.

The wmt17 dataset is available at https://huggingface.co/datasets/wmt/wmt17.

### Supplemental Information

Supplemental information for this article can be found online at http://dx.doi.org/10.7717/peerj-cs.3199#supplemental-information.

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
