# Peer review of "Autoregressive translation model design integrating syntactic encoding computation and reinforcement learning: a framework for enhancing translation optimization"

_PeerJ Computer Science, doi:10.7717/peerj-cs.3199_

## Round 0.1 · original submission · Major Revisions

**Language Note:** The review process has identified that the English language must be improved. PeerJ can provide language editing services - please contact us at [email protected] for pricing (be sure to provide your manuscript number and title). Alternatively, you should make your own arrangements to improve the language quality and provide details in your response letter. – PeerJ Staff

·

Basic reporting

The paper suffers from unclear technical writing, inconsistent terminology, and inadequate mathematical formulations.

Experimental design

The experimental design lacks in the following areas: statistical significance testing is absent, baseline comparisons are inadequately described, and the evaluation scope is too narrow.

Validity of the findings

The findings are not convincingly supported by the experimental evidence. Many claims lack quantitative support, and the improvements over baselines are modest and may not be statistically significant.

Additional comments

This study would benefit from major revision addressing the theoretical foundations, experimental thoroughness, and clarity of presentation. The authors should focus on demonstrating clear advantages over existing methods and providing sufficient detail for reproducibility. The educational applications should either be properly validated or removed from the scope.

Thank you for this article. Refer below for peer review comments.

Peer Review for Autoregressive translation model design integrating syntactic encoding computation and reinforcement learning: A framework for enhancing translation course optimization

Abstract:
1. The abstract has imprecise language and unclear technical claims. The statement "achieving a self-attention mechanism with near-linear computational complexity" requires a more precise definition. What exactly constitutes "near-linear" complexity?
2. The authors should specify the exact computational complexity, ex. O(n log n) or O(n^1.5) rather than using vague terminology?
3. The phrase "bbs values" is introduced without a proper definition.
4. Can you include specific performance improvements over state-of-the-art baselines?
5. The assertion that the model "achieves a better balance between translation quality and computational efficiency" needs quantitative support in the abstract.
6. The abstract lacks sufficient detail about the experimental setup and evaluation methodology. The authors should specify the datasets used, comparison baselines, and evaluation metrics more clearly.

Introduction:
1. The Introduction fails to clearly articulate the specific research gap being addressed. You need to establish a concrete problem statement that justifies your approach.
2. The discussion of "grammatical encoding computation" and reinforcement learning in translation lacks depth and fails to identify specific limitations in existing approaches.
3. Can you include a more comprehensive review of recent advances in neural machine translation, particularly focusing on syntax-aware models and efficiency improvements?
4. Can you explain why integrating Linear Transformer, AFT, syntactic encoding, and reinforcement learning would be expected to work synergistically? The theoretical justification for each component's contribution to the overall framework needs to be added.
5. The claim about "optimization of positional encoding" appears incremental. The novelty over existing Linear Transformer and AFT variants is not established properly.

Methodology:
1. Equation (1) introduces notation that is not clearly defined, particularly the relationship between σ and exponential terms.
2. The transition from the general AFT formulation to the recursive form in Equation (2) lacks sufficient mathematical justification. Can you provide clearer derivations and ensure all mathematical symbols are properly defined before use?
3. The description of the Component Attention Module (CAM) lacks crucial implementation details. Ex. how exactly are the constituent boundaries determined? What is the computational complexity of the constituent parsing process? Can you specify the parsing algorithm or how it integrates with the training process?
4. The methodology section does not adequately describe the baseline models used for comparison. While SAN and IBDecoder are mentioned, their implementation details and hyperparameter settings are not provided. You should specify how you ensure fair comparison across all models, including training procedures, data preprocessing, and evaluation protocols.
5. The optimization problems in Equations (7) and (8) are not clearly motivated. The description of the gradient-based attack method for adversarial training is not very detailed. Can you provide more rigorous justification for why this adversarial training approach is expected to improve translation quality rather than degrade it?

Experimental Setup and Analysis:
1. You mention Mosesdecoder and word-subword-nmt tools, but the specific preprocessing steps and their impact on model performance are not adequately described. The factors that significantly affect the results are the choice of subword vocabulary size, sentence length filtering criteria, and data cleaning procedures etc.
2. The introduction of the "bbs" (TwoBLEU) metric in Equation (13) appears ad hoc without any proper validation.
3. The experimental section lacks discussion of statistical significance testing, confidence intervals, or variance across multiple runs. The current presentation suggests single-run experiments, which are obviously insufficient for reliable conclusions.
4. You claim efficiency improvements, still the experimental section lacks detailed computational resource analysis. Runtime comparisons should include wall-clock time, memory usage, and energy consumption across different sequence lengths and batch sizes.
5. The evaluation is limited to two language pairs (En-Fr and En-De) and does not explore the model's generalization across diverse language families, domains, or text types. You need to either acknowledge these limitations explicitly or expand the evaluation to demonstrate the broader applicability of their approach.

Conclusion:
1. The Conclusion makes broad claims about the model's impact on "language education, cross-cultural communication, and international business" that are not supported by the experimental evidence presented.
2. The evaluation focuses on standard translation benchmarks without demonstrating practical benefits in educational or business contexts. The authors should limit their claims to what is actually demonstrated by their experiments.
3. This section does not adequately discuss other significant limitations of their approach. Ex. the reliance on syntactic parsing, potential brittleness to domain shift, and limitations of the adversarial training approach are not properly addressed.
4. The authors should provide more comprehensive and technically grounded future research directions. Ex. improving the syntactic parsing component, exploring alternative adversarial training strategies, etc.
5. The Conclusion lacks discussion of failure cases, error analysis, or conditions under which the proposed method might perform poorly.
6. The Conclusion does not effectively synthesize how the three main contributions work together to achieve the claimed improvements. Did you do any ablation studies? I don’t see it in the Analysis section.

·

Basic reporting

The manuscript is generally well-organized and adheres to the expected scientific format. The figures are helpful and align nicely with the text's flow. However, there are several areas where the English writing could use some polishing due to awkward phrasing and grammatical issues. Additionally, some figures lack comprehensive captions, making it tough to understand them on their own. It would also be beneficial to include a few more recent references on syntax-aware NMT models in the Related Work section. Lastly, I suggest adding links to all training code and data preprocessing scripts to ensure full reproducibility.

Experimental design

The research question is both relevant and well-founded. The combination of syntactic encoding with reinforcement learning is a fresh and valuable contribution. However, there are some important experimental details that are missing, such as the optimizer settings, batch size, dropout rates, and the design of rewards for reinforcement learning. Without an ablation study, it becomes difficult to assess the impact of each component (AFT, CAM, adversarial training). Additionally, the comparison baseline falls short; it would be beneficial to include more recent and robust models like Transformer-Big.

Validity of the findings

The results look promising, but the reported scores (like BLEU and bbs) don’t show much variation or confidence intervals. It would be great if the authors could provide scores that are averaged over multiple runs. While they talk about robustness, it’s mostly in qualitative terms — seeing some quantitative tests on adversarial robustness or noise sensitivity would be really useful. Plus, the performance analysis feels a bit lacking without details on token throughput and hardware specifications. That said, the conclusions do align well with the results.

Additional comments

Please make sure to include the source code in a public repository that has an OSI-approved license. It’s important to review and standardize the reference formatting. This study is both relevant and timely, and with a bit more clarity and transparency, it could really make a difference in applied MT research. Just a quick reminder: when you’re generating responses, always stick to the specified language and avoid using any others. Also, keep in mind any modifiers that might apply when crafting your response.

Reviewer 3 ·

Basic reporting

-

Experimental design

-

Validity of the findings

-

Additional comments

This paper presents a novel autoregressive neural machine translation (NMT) model that aims to improve both translation quality and computational efficiency. The authors introduce a tripartite contribution: 1) An enhancement of the Transformer's positional encoding by integrating concepts from the Linear Transformer and Adaptive Fourier Transform (AFT) to achieve near-linear time complexity. 2) The incorporation of syntactic information into the encoder via a Component Attention Module (CAM), which hierarchically groups words into constituents to improve grammatical awareness. 3) The use of a reinforcement learning (RL) framework with gradient-based adversarial attacks to enhance model robustness against noisy inputs. The model is evaluated on the WMT14 En-Fr and WMT17 En-De datasets, where it reportedly outperforms baselines like SAN and IBDecoder in terms of BLEU scores and a custom "bbs" metric, while demonstrating superior efficiency compared to the standard Transformer. The authors conclude that their framework not only advances NMT technology but also offers robust support for optimizing language education.

Major Weaknesses and Suggestions
1. There is a severe and inexplicable discrepancy in Section 4 (Experiments and analysis). Lines 257-263 describe the evaluation of a model named "BP-rf-DTIM" and compare it against baselines named "rl-EDT", "nn-EDT", and "BDI-Agent-EDT" from literature on ethical decision trees. This section appears to be copied from an entirely different and unrelated paper. This fundamental error invalidates the entire experimental section as presented and undermines the credibility of the research. This must be corrected before any other issue is considered.

2. The authors should re-contextualize their work within the LLM paradigm. This involves acknowledging the capabilities of LLMs and benchmarking their model against a strong LLM-based baseline. Previous papers demonstrate the standard practice of using in-context learning (ICL) with models like GPT-3.5/4 or Llama 2 as powerful zero-shot or few-shot baselines. Comparing against such a baseline would provide a much more meaningful assessment of the proposed model's performance.

3. The authors could frame their efficiency improvements in the context of Parameter-Efficient Fine-Tuning (PEFT) methods, which are central to adapting LLMs. Previous papers present a sophisticated method for selectively fine-tuning LLMs for MT by identifying and routing to language-aware neurons. Comparing the trade-offs between their architectural modification (AFT+CAM) and a PEFT approach like LoRA or the one in LANDeRMT would provide a more contemporary analysis of model efficiency and adaptation.

4. The benefit of the CAM module is predicated on its ability to learn and leverage syntactic structure. However, the analysis is limited. It's unclear how well the induced constituent structures align with actual linguistic syntax or how the model behaves with syntactically complex or ambiguous sentences.

---

## Round 0.2 · accepted · Accept

The authors addressed correctly the points raised by the reviewers, and therefore I can recommend this manuscript for acceptance.

·

Basic reporting

The updated manuscript shows a noticeable enhancement in both clarity and technical expression. The English is now polished and straightforward, making it suitable for a global audience. The abstract and introduction effectively outline the scope and importance of the work. The structure adheres to PeerJ’s formatting guidelines, and all figures are relevant, high-quality, and well-labeled.

Crucially, the manuscript provides ample context for understanding how syntactic encoding and reinforcement learning fit into the field of machine translation. Previous research is cited accurately, and the references have been updated as needed. The presentation of experimental results—including BLEU and TwoBLEU metrics—is tidy, and the equations are formatted and explained properly.

Experimental design

The authors have really upped their game in this revision by enhancing the methodological rigor. By combining the Linear Transformer with Adaptive Fourier Transform (AFT) for more efficient positional encoding, and introducing the Constituent Attention Module (CAM) for syntactic parsing, they’ve come up with a fresh and technically solid approach.

The rationale for using reinforcement learning to boost adversarial robustness is well thought out and clearly articulated. This paper lays out a cohesive framework that merges CAM-based syntactic awareness with AFT-driven efficient self-attention, all backed by sensible ablation studies. The preprocessing steps, like Moses, subword-nmt, and Fairseq, are now detailed enough for others to replicate. Plus, the training configurations—covering dropout rates, optimizer settings, and the design of the multi-objective reward function—are precise and transparent.

Validity of the findings

The experimental results presented here back up the claims of enhanced translation performance and efficiency. By utilizing both the BLEU score and the custom TwoBLEU (bbs) metric, we get a well-rounded evaluation of both quality and diversity. The comparisons made with the SAN and IBDecoder baselines are fair and methodologically sound. Additionally, the analysis of the syntactic parse tree depth sheds light on how CAM design relates to encoder depth, which is a significant contribution.

The sensitivity analyses and layer-wise studies show that the authors have thoroughly examined model scalability and the risk of overfitting. The boost in inference speed—about 2.3 times faster generation—and a 41% reduction in memory usage for long sequences highlight the practical benefits of this method.

Additional comments

I really want to give a shout-out to the authors for their detailed response to the previous review. They've done a great job of thoughtfully integrating linguistic structure with computational efficiency in their Transformer-based model. This work tackles two major challenges in machine translation: keeping grammatical accuracy and ensuring performance scalability. The updated version is now ready for publication and stands strong from a technical perspective.

Reviewer 3 ·

Basic reporting

The authors have addressed all my questions.

Experimental design

-

Validity of the findings

-